# Generative Neurosymbolic Machines

**Jindong Jiang**
Department of Computer Science
Rutgers University
jindong.jiang@rutgers.edu

**Sungjin Ahn**
Department of Computer Science
Rutgers University
sjn.ahn@gmail.com

## Abstract

Reconciling symbolic and distributed representations is a crucial challenge that can potentially resolve the limitations of current deep learning. Remarkable advances in this direction have been achieved recently via generative object-centric representation models. While learning a recognition model that infers object-centric symbolic representations like bounding boxes from raw images in an unsupervised way, no such model can provide another important ability of a generative model, i.e., generating (sampling) according to the structure of learned world density. In this paper, we propose Generative Neurosymbolic Machines, a generative model that combines the benefits of distributed and symbolic representations to support both structured representations of symbolic components and density-based generation. These two crucial properties are achieved by a two-layer latent hierarchy with the global distributed latent for flexible density modeling and the structured symbolic latent map. To increase the model flexibility in this hierarchical structure, we also propose the StructDRAW prior. In experiments, we show that the proposed model significantly outperforms the previous structured representation models as well as the state-of-the-art non-structured generative models in terms of both structure accuracy and image generation quality.

## 1 Introduction

Two central abilities in human and machine intelligence are to learn abstract representations of the world and to generate imaginations in such a way to reflect the causal structure of the world. Deep latent variable models like variational autoencoders (VAEs) [28, 36] offer an elegant probabilistic framework to learn both these abilities in an unsupervised and end-to-end trainable fashion. However, the single distributed vector representations used in most VAEs provide in practice only a weak or implicit form of structure induced by the independence prior. Therefore, in representing complex, high-dimensional, and structured observations such as a scene image containing various objects, the representation is rather difficult to express useful structural properties such as modularity, compositionality, and interpretability. These properties, however, are believed to be crucial in resolving limitations of current deep learning in various System 2 [26] related abilities such as reasoning [4], causal learning [37, 34], accountability [11], and systematic out-of-distribution generalization [2, 42].

There have been remarkable recent advances in resolving this challenge by learning to represent an observation as a composition of its entity representations, particularly in an object-centric fashion for scene images [13, 29, 16, 41, 6, 15, 12, 10, 30, 9, 24, 44]. Equipped with more explicit inductive biases such as spatial locality of objects, symbolic representations, and compositional scene modeling, these models provide a way to recognize and generate a given observation via the composition of interacting entity-based representations. However, most of these models do not support the other crucial ability of a generative model: generating imaginary observations by learning the density of the observed data. Although this ability to imagine according to the density of the possible worlds plays a crucial role, e.g., in world models required for planning and model-based reinforcement

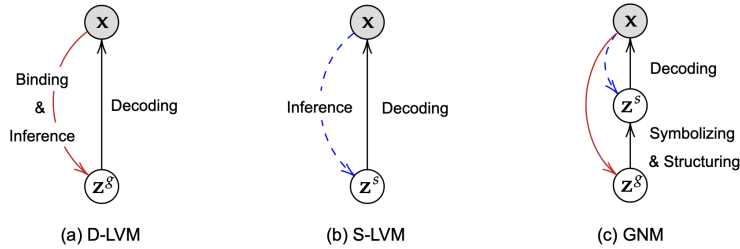

**Figure 1:** Graphical models of D-LVM, S-LVM, and GNM. $\mathbf{z}^g$ is the global distributed latent representation, $\mathbf{z}^s$ is the symbolic structured representation, and $\mathbf{x}$ is an observation. The red solid-arrow-line indicates joint learning of variable binding and value inference and the blue dotted-arrow-line indicates only value inference.

learning [20, 19, 1, 33, 22, 35, 21], most previous entity-based models can only synthesize artificial images by manually configuring the representation, but not according to the underlying observation density. Although this ability is supported in VAEs [28, 17], lacking an explicitly compositional structure in its representation, it easily loses in practice the global structure consistency when generating complex images [40, 17].

In this paper, we propose Generative Neurosymbolic Machines (GNM), a probabilistic generative model that combines the best of both worlds by supporting both symbolic entity-based representations and distributed representations. The model thus can represent an observation with symbolic compositionality and also generate observations according to the underlying density. We achieve these two crucial properties simultaneously in GNM via a two-layer latent hierarchy: the top layer generates the global distributed latent representation for flexible density modeling and the bottom layer yields from the global latent the latent structure map for entity-based and symbolic representations. Furthermore, we propose StructDRAW, an autoregressive prior supporting structured feature-drawing to improve the expressiveness of latent structure maps. In experiments, we show that for both the structure accuracy and image clarity, the proposed model significantly outperforms the previous structured representation models as well as highly-expressive non-structured generative models.

## 2 Symbolic and Distributed Representations in Latent Variable Models

**Variable binding and value inference.** What functions are a representation learning model based on an autoencoder (e.g., VAE) performing? To answer this, we provide a perspective that separates the function of the encoder into two: variable binding and value inference. *Variable binding (or grounding)* is to assign a specific role to a variable (or a group of variables) in the representation vector. For instance, in VAEs each variable in the latent vector is encouraged to have its own meaning through the independence prior. In an ideal case with perfect disentanglement, we would expect to find a specific variable in the latent vector that is in charge of controlling $x$-coordinate of an object in an image [23, 8, 27, 31]. That is, the variable is grounded on the object's position. However, in practice, such perfect disentanglement is difficult to achieve [31], and thus the representation shows correlations among the values in it. *Value inference* is to assign a specific *value* to the binded variable, e.g., in our example, a coordinate value. In VAE, the variable binding is fixed after training—the same variable represents the same semantics for different inputs—but the inferred value can be changed per observation (e.g., if the object position changes). In VAEs, both variable binding and value inference are learned jointly.

**Distributed vs. Symbolic Representations.** We define a symbolic representation as a latent variable to which a semantic role is solely assigned independently to other variables. For example, in object-centric latent variable models [13, 30, 10, 24], a univariate Gaussian distribution $p(z_x^{\text{where}}) = \mathcal{N}(\mu_x, \sigma_x)$ can be introduced to define a symbolic prior on the $x$-coordinate of an object in an image. (Then, the final $x$-coordinate can be computed by $I_x \times \text{sigmoid}(z_x^{\text{where}})$ with $I_x$ the image width.) On the contrary, in distributed representations, variable binding can be distributed. That is, a semantic variable can be represented in a distributed way across the whole latent vector with correlation among the vector elements. A single Gaussian latent vector of the standard VAE is a representative example. Although VAEs objective encourages the disentanglement of each variable, it is, in general, more difficult to achieve such complete disentanglement than symbolic representations.

Distributed latent variable models (D-LVM) in general provides more flexibility than symbolic latent variable models (S-LVM) as the variable binding can be distributed and, more importantly, learned from data. This learnable binding allows turning the prior latent distribution into the distribution

of complex high-dimension observations. In S-LVMs, such flexibility can be significantly limited to representing the semantics of the fixed and interpretable binding. For instance, if we introduce a prior on a symbolic variable representing the number or positions of objects in an image but in a way that does not match the actual data distribution, S-LVMs cannot fix this to generate according to the observed data distribution. However, S-LVM brings various advantages that are, in general, more difficult to be achieved in D-LVMs. The completely disentangled symbols facilitate interpretability, reasoning, modularity, and compositionality. Also, since the encoder only needs to learn value inference, learning can be facilitated. See Fig. 1 (a)-(c) for an illustration.

**Object-Centric Representation Learning.** There are two main approaches to this. The bounding-box models [13, 29, 10, 30, 9, 24] infer object appearances along with their bounding boxes and reconstruct the image by placing objects according to their bounding boxes. Scene-mixture models [16, 41, 6, 15, 12, 44, 32] try to *partition* the image into several layers of images, potentially one per object, and reconstruct the full image as a pixel-wise mixture of these layered images. The bounding-box models utilize many symbolic representations such as the number of objects, and positions and sizes of the bounding boxes. Thus, while entertaining various benefits of symbolic representation, it also inherits the above-mentioned limitations, and thus currently no bounding-box model can generate according to the density of the data. Scene-mixture models such as [16, 15, 6, 32] rely less on symbolic representations as each mixture component of a scene is generated from a distributed representation. However, these models also do not support the density-based generation as the mixture components are usually independent of each other. Although GENESIS [12] has an autoregressive prior on the mixture components and thus can support the density-aware generation in principle, our experiment results indicate limitations of the approach.

# 3 Generative Neurosymbolic Machines

## 3.1 Generation

We formulate the generative process of the proposed model as a simple two-layer hierarchical latent variable model. In the top layer, we generate a distributed representation $\mathbf{z}^g$ from the global prior $p(\mathbf{z}^g)$ to capture the global structure with the flexibility of the distributed representation. From this, the structured latent representation $\mathbf{z}^s$ containing symbolic representations is generated in the next layer using the structuring prior $p(\mathbf{z}^s|\mathbf{z}^g)$. The observation is constructed from the structured representation using the rendering model $p(\mathbf{x}|\mathbf{z}^s)$. With $\mathbf{z} = \{\mathbf{z}^g, \mathbf{z}^s\}$, we can write this as

$$p_\theta(\mathbf{x}) = \int p_\theta(\mathbf{x}|\mathbf{z}^s)p_\theta(\mathbf{z}^s|\mathbf{z}^g)p_\theta(\mathbf{z}^g)\mathrm{d}\mathbf{z} . \tag{1}$$

**Global Representation**. The global representation $\mathbf{z}^g$ provides the flexibility of the distributed representation. That is because the meaning of a representation vector is distributed and not predefined but endowed later by *learning* from the data, it allows complex distributions (e.g., highly multimodal and correlated distribution on the number of objects and their positions in a scene) to be modeled with the representation. In this way, the global representation $\mathbf{z}^g$ contains an abstract and flexible summary necessary to generate the observation but lacks an explicit compositional and interpretable structure. Importantly, the role of the global representation in our model is different from that in VAE. Instead of directly generating the observation from this distributed representation by having $p(\mathbf{x}|\mathbf{z}^g)$, it acts as high-level abstraction serving for constructing a structured representation $\mathbf{z}^s$, called the *latent structure map*, via the structuring model $p_\theta(\mathbf{z}^s|\mathbf{z}^g)$. A simple choice for the global representation is a multivariate Gaussian distribution $\mathcal{N}(\mathbf{0}, \mathbf{1}_{d_g})$.

**Structured Representation.** In the latent structure map, variables are explicitly and completely disentangled into a set of components. To obtain this in the image domain, we first build from the global representation $\mathbf{z}^g$ a feature map $\mathbf{f}$ of $(H \times W \times d_f)$-dimension with $H$ and $W$ being the spatial dimension and $d_f$ being the feature dimension in each spatial position. Thus, $H$ and $W$ are hyperparameters controlling the maximum number of components and usually a much smaller number (e.g., $4 \times 4$) than the image resolution. Then, for each feature vector $\mathbf{f}_{hw}$, a component latent $\mathbf{z}_{hw}^s$ of the latent structure map is inferred. Depending on applications, $\mathbf{z}_{hw}^s$ can be a set of purely symbolic representations or a hybrid of symbolic and distributed representations.

For multi-object scene modeling, which is our main application, we use a hybrid representation $\mathbf{z}_{hw}^s = [\mathbf{z}_{hw}^{\text{pres}}, \mathbf{z}_{hw}^{\text{where}}, \mathbf{z}_{hw}^{\text{depth}}, \mathbf{z}_{hw}^{\text{what}},]$ to represent the presence, position, depth, and appearance of a

component, respectively. Here, appearance $\mathbf{z}_{hw}^{\text{what}}$ is a distributed representation while the others are symbolic. We use Bernoulli distributions for presence and Gaussian distributions for the others. We also introduce the background component $\mathbf{z}^b$, which represents a part of the observation remained after the explanation by the other foreground components. We can consider the background as a special foreground component for which we only need to learn the appearance while fixing the other variables constant. Then, we can write the structuring model as follows:

$$p_\theta(\mathbf{z}^s \,|\, \mathbf{z}^g) = p_\theta(\mathbf{z}^b \,|\, \mathbf{z}^g) \prod_{h=1}^{H} \prod_{w=1}^{W} p_\theta(\mathbf{z}_{hw}^s \,|\, \mathbf{z}^g) \,, \tag{2}$$

where $\mathbf{z}^s = \mathbf{z}^b \cup \{\mathbf{z}_{hw}^s\}$ and $p_\theta(\mathbf{z}_{hw}^s | \mathbf{z}^g) = p_\theta(\mathbf{z}_{hw}^{\text{pres}} | \mathbf{z}^g) p_\theta(\mathbf{z}_{hw}^{\text{what}} | \mathbf{z}^g) p_\theta(\mathbf{z}_{hw}^{\text{where}} | \mathbf{z}^g) p_\theta(\mathbf{z}_{hw}^{\text{depth}} | \mathbf{z}^g)$.

The latent structure map might look similar to that in SPACE [30]. However, in SPACE, independent symbolic priors are used to obtain scalability, and thus it cannot model the underlying density. Unlike SPACE, the proposed model generates the latent structure map from the global representation, which is distributed and groundable (binding learnable). This is crucial because by doing so, we achieve both flexible density modeling and benefits of symbolic representations. Unlike the other models [13, 6, 12], this approach is also efficient and stable for object-crowded scenes [30, 9, 24].

**Renderer.** Our model adopts the typical renderer module $p(\mathbf{x}|\mathbf{z}^s)$ used in bounding-box models, e.g., SPACE. We provide the implementation details of the renderer in Appendix.

### 3.2 StructDRAW

One limitation of the above model is that the simple Gaussian prior for $p(\mathbf{z}^g)$ may not have enough flexibility to express complex global structures of the observation, a well-known problem in VAE literature [40, 17]. One way to resolve this problem is to generate the image autoregressively at pixel-level [40], or by superimposing several autoregressively-generated sketches on a canvas [17, 18]. However, these approaches cannot be adopted in GNM as they generate images directly from the global latent without structured representation.

In GNM, we propose StructDRAW to make the global representation express complex global structures when it generates the latent structure map. The overall architecture of StructDRAW, illustrated in Appendix, basically follows that of ConvDRAW [17] but with two major differences. First, unlike other ConvDRAW models [14, 19, 17], StructDRAW *draws not pixels but an abstract structure on feature space*, i.e., the latent feature map, by $\mathbf{f} = \sum_{\ell=1}^{L} \mathbf{f}_\ell$ with $\ell$ being the autoregressive step index. This abstract map has a much lower resolution to be drawn than the pixel-level drawing, and thus *can focus more effectively on drawing the structure instead of the pixel drawing*. Pixel-level drawing is passed on to the component-wise renderer that composites the full observation by rendering each component $\mathbf{z}_{hw}^s$ individually.

Second, to encourage full interaction among the abstract components, we introduce an interaction layer before generating latent $\mathbf{z}_\ell^g$ at each $\ell$-th StructDRAW step. The global correlation is important, especially if the image is large. However, in ConvDRAW, such interaction can happen only locally via convolution and successive autoregressive steps of such local interactions, potentially missing the global long-range interaction. To this end, in our implementation, we found that a simple approach of using a multilayer perceptron (MLP) layer as the full interaction module works well. However, it is also possible to employ other interaction models, such as the Transformers [43] or graph neural networks [3]. Autoregressive drawing has also been used in other object-centric models [13, 6, 12]. However, unlike these models, the number of drawing steps in GNM is not tied to the number of components. Thus, GNM is scalable to object-crowded scenes. In our experiments, only 4 StructDRAW-steps were enough to model 10-component scenes, while other autoregressive models require at least 10 steps.

### 3.3 Inference

For inference, we approximate the intractable posterior by the following mean-field decomposition:

$$p_\theta(\mathbf{z}^g, \mathbf{z}^s \,|\, \mathbf{x}) \approx q_\phi(\mathbf{z}^g \,|\, \mathbf{x}) q_\phi(\mathbf{z}^b \,|\, \mathbf{x}) \prod_{h=1}^{H} \prod_{w=1}^{W} q_\phi(\mathbf{z}_{hw}^s \,|\, \mathbf{x}) \,. \tag{3}$$

As shown, our model provides *dual representations* for an observation $\mathbf{x}$. That is, the global latent $\mathbf{z}^g$ represents the scene as a flexible distributed representation, and the structured latents $\mathbf{z}^s$ provides a structured symbolic representation of the same observation.

**Image Encoder.** As all modules take $\mathbf{x}$ as input, we share an image encoder $f_{\text{enc}}$ across the modules. The encoder is a CNN yielding an intermediate feature map $\mathbf{f}^x = f_{\text{enc}}(\mathbf{x})$.

**Component Encoding.** The component encoder $q_\phi(\mathbf{z}_{hw}^s|\mathbf{x})$ takes the feature map $\mathbf{f}^x$ as input to generate the background and the component latents in a similar way as done in SPACE except that the background is not partitioned. We found that conditioning the foreground $\mathbf{z}^s$ on the background $\mathbf{z}^b$ (or vice versa) does not help much because if both modules are learned simultaneously from scratch, one module can dominantly explain $\mathbf{x}$ and weaken the training of the other module. To resolve this, we found curriculum training to be effective (described in a following section.)

**Global Encoding.** For GNM with the Gaussian global prior, the global encoding is the same as VAE. However, to use StructDRAW prior, we use an autoregressive model: $q_\phi(\mathbf{z}^g|\mathbf{x}) = \prod_{\ell=1}^{L} q_\phi(\mathbf{z}_\ell|\mathbf{z}_{<\ell}, \mathbf{x})$ to generate the feature map $\mathbf{f} = \sum_{\ell=1,\dots,L} \text{CNN}(\mathbf{h}_{\text{dec},\ell})$. The feature map $\mathbf{h}_{\text{dec},\ell}$ drawn at the $\ell$-th step is generated by the following steps: (1) $\mathbf{h}_{\text{enc},\ell} = \text{LSTM}_{\text{enc}}(\mathbf{h}_{\text{enc},\ell-1}, \mathbf{h}_{\text{dec},\ell-1}, \mathbf{f}^x, \mathbf{f}_{\ell-1})$, (2) $\boldsymbol{\mu}_\ell, \boldsymbol{\sigma}_\ell = \text{MLP}_{\text{interaction}}(\mathbf{h}_{\text{enc},\ell})$, (3) $\mathbf{z}_\ell \sim \mathcal{N}(\boldsymbol{\mu}_\ell, \boldsymbol{\sigma}_\ell)$, and (4) $\mathbf{h}_{\text{dec},\ell} = \text{LSTM}_{\text{enc}}(\mathbf{z}_\ell, \mathbf{h}_{\text{dec},\ell-1}, \mathbf{f}_{\ell-1})$. Here, $\mathbf{f}_\ell = \sum_{l=1}^{\ell} \text{CNN}(\mathbf{h}_{\text{dec},l})$.

### 3.4 Learning

We train the model by optimizing the following Evidence Lower Bound (ELBO): $\mathcal{L}_{\text{ELBO}}(\mathbf{x}; \theta, \phi) =$

$$\mathbb{E}_{q_\phi(\mathbf{z}^s|\mathbf{x})}\left[\log p_\theta(\mathbf{x} \mid \mathbf{z}^s)\right] - D_{\text{KL}}\left[q_\phi(\mathbf{z}^g \mid \mathbf{x}) \parallel p_\theta(\mathbf{z}^g)\right] - D_{\text{KL}}\left[q_\phi(\mathbf{z}^s \mid \mathbf{x}) \parallel p_\theta(\mathbf{z}^s \mid \mathbf{z}^g)\right] . \quad (4)$$

where $D_{\text{KL}}(q \parallel p)$ is Kullback-Leibler Divergence. For the latent structure maps, as an auxiliary term, we also add standard KL terms between the posterior and *unconditional* prior such as $D_{\text{KL}}\left[q_\phi(\mathbf{z}^{\text{pres}} \mid \mathbf{x}) \| \text{Ber}(\rho)\right]$ and $D_{\text{KL}}\left[q_\phi(\mathbf{z}^b \mid \mathbf{x}) \parallel \mathcal{N}(\mathbf{0}, \mathbf{1})\right]$. This allows us to impose prior knowledge to the learned posteriors [7]. See the supplementary material for detailed equations for this auxiliary loss. We apply curriculum training to deal with the racing condition between the background and component modules, both trying to explain the full observation. For this, we suppress the learning of the background network in the early training steps and give a preference to the foreground modules to explain the scene. When we begin to fully train the background, it focuses on the background.

## 4 Experiments

**Goals and Datasets.** The goals of the experiments are (i) to evaluate the quality and properties of the generated images in terms of clarity and scene structure, (ii) to understand the factors of the datasets and hyperparameters that affect the performance, and (iii) to perform ablation studies to understand the key factors in the proposed architecture. We use the following three datasets:

*MNIST-4.* In this dataset, an image is partitioned into four areas (top-right, top-left, bottom-right, and bottom-left), and one MNIST digit is placed in each quadrant. To make structural dependency among these components, we generated the images as follows. First, a random digit of a class randomly sampled between 0 and 6 is generated in a random position in the top-left quadrant. Then, starting from the top-left, a random digit is placed to each of the other quadrants with the digit class increased by one in the clockwise direction. The positions of these digits are symmetric to each other on the $x$-axis and $y$-axis whose origin is the center of the image. See Fig. 2 for examples.

*MNIST-10.* To evaluate the effect of the number of components and complexity of the dependency structure, we also created a similar dataset containing ten MNIST-digits and a more complex dependency structure. The images are generated as follows. An image is also split into four quadrants. For each quadrant, four mutually exclusive sets of digit classes are assigned: $Q_1 = \{0, 1\}$, $Q_2 = \{2, 3, 4\}$, $Q_3 = \{8, 9\}$, and $Q_4 = \{5, 6, 7\}$ in the clock-wise order from top-left quadrant ($Q_1$), respectively. Then, the following structural conditions are applied. $Q_1$ and $Q_3$ are placed randomly and at the same within-quadrant position. Digits in $Q_2$ and $Q_4$ have no position dependency and are placed randomly within the quadrants. To impose a stochastic dependency, the quadrants are diagonally swapped at random.

*Arrow Room.* This dataset contains four 3D objects in a 3D space similar to CLEVR [25]. The objects are combinatorially generated from 8 colors, 4 shapes, and 2 material types. Among the four

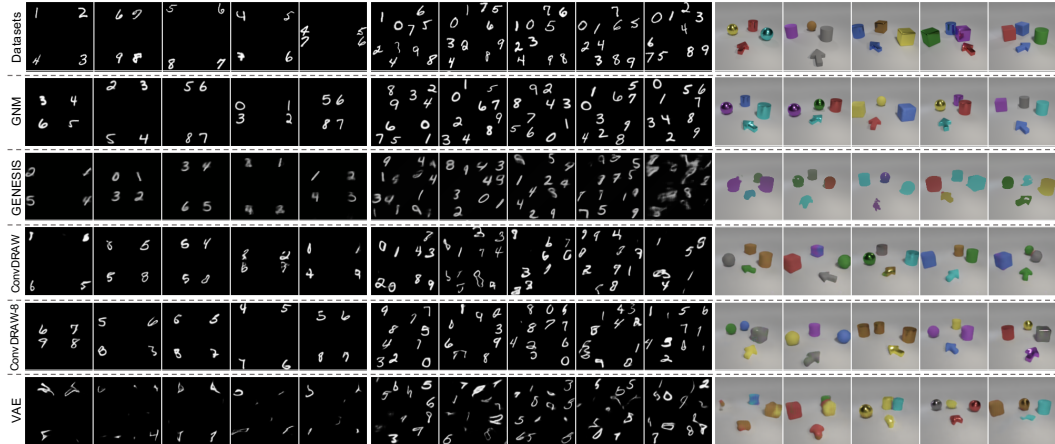

**Figure 2:** Datasets and generation examples. MNIST-4 (left), MNIST-10 (middle), and Arrow room (right)

**Table 1:** Quantitative results on scene structure accuracy, discriminability test, and log-likelihood.

| Dataset | ARROW | | | MNIST-10 | | | MNIST-4 | | |
|---|---|---|---|---|---|---|---|---|---|
| Metrics | S-Acc | D-Steps | LL | S-Acc | D-Steps | LL | S-Acc | D-Steps | LL |
| GNM | **0.976** | **11099** | **33809** | **0.824** | **2760** | 10450 | **0.984** | **3920** | 10964 |
| GENESIS | 0.092 | 1900 | 33241 | 0.000 | 160 | 9560 | 0.296 | 200 | 10496 |
| ConvDRAW | 0.176 | 3800 | 33740 | 0.000 | 1200 | 10544 | 0.048 | 2400 | 11020 |
| ConvDRAW-8 | 0.420 | 3499 | 33749 | 0.000 | 1680 | **10590** | 0.604 | 3440 | **11036** |
| VAE | 0.036 | 5499 | 33672 | 0.000 | 279 | 10031 | 0.000 | 319 | 10895 |

objects, one always has the arrow shape, two other objects always have the same shape, and the last one, which the arrow always points to, has a unique shape. Object colors are randomly sampled, but the same material is applied to all objects within an image. The arrow is the closest to the camera.

**Baselines.** We compare GNM to the following baselines. (i) GENESIS is the main baseline which, like GNM, is supposed to support both structured representation and density-based generation. (ii) ConvDRAW is one of the most powerful VAE models that focuses on density-based generation without the burden of learning structured representation. Here we want to investigate whether GNM can match or outperform ConvDRAW even while simultaneously learning a structured representation. Finally, (iii) VAE is a model representing the no-structure and no-autoregressive-prior case. We set the default drawing steps of GNM and ConvDRAW to 4 but also tested with 8 steps.

**Evaluation Metrics.** We use three metrics to evaluate the performance of our model. For the (i) scene structure accuracy (S-Acc), we manually classified the 250 generated images per model into `success` or `failure` based on the correctness of the scene structure in the image without considering generation quality. When we cannot recognize the digit class, however, we also labeled those images as `failures`. For the (ii) discriminability score (D-Steps), we measure how difficult it is for a binary classifier to discriminate the generated images from the real images. This metric considers both the image clarity and dependency structure because a more realistic image, i.e., satisfying both of these criteria, should be more difficult to discriminate, i.e., it takes more time for the binary classifier to converge. For this metric, we measure the number of training steps required for the binary classifier to reach 90% classification accuracy. Finally, we estimated the (iii) log-likelihood (LL) using importance sampling with 100 posterior samples [5].

## 4.1 Results

**Qualitative Analysis of Samples.** In Figure 2, we show the samples from the compared models. We first see that *the GNM samples are almost impossible to distinguish from the real images.* The image is not only clear but also has proper scene structure following the constraints in the dataset generation. GENESIS generates blurry and unrecognizable digits, and the structure is not correct in many scenes. For the ARROW dataset, we see that the generation is oversimplified and does not model the metal texture. The shape is also significantly distorted by lighting. For ConvDRAW, many digits look different from the real and sometimes unrecognizable, and many scenes with incorrect

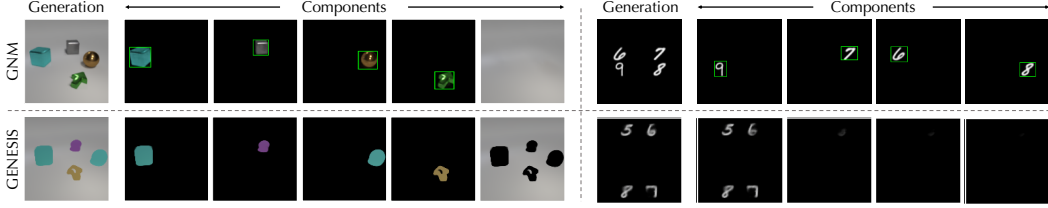

**Figure 3:** Component-wise generation with GNM and GENESIS. Green bounding boxes represents $\mathbf{z}^{\text{where}}$.

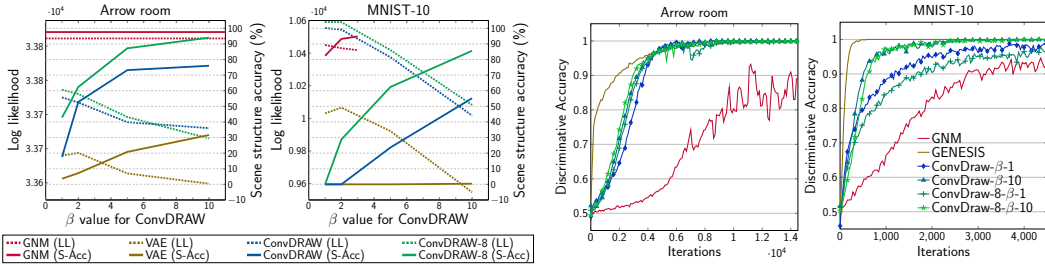

**Figure 4:** Beta effect (left) and learning curve for binary discriminator (right).

structures are also observed. For the ARROW dataset, object colors are sometimes not consistent, and the arrow directs the wrong object. We can also see a scene where all objects have different shapes not existing in the real dataset. Finally, the VAE samples are significantly worse than the other models. In Figure 3, we also compare the decomposition structure between GNM and GENESIS. It is interesting to see that GENESIS cannot decompose objects with the same color. Not surprisingly, VAE with neither the autoregressive drawing prior nor the structured representation performs the worst. See supplementary for more generation results and different effects on $\mathbf{z}^{\text{g}}$ and $\mathbf{z}^{\text{s}}$ sampling.

**Scene Structure Accuracy.** For quantitative analysis, we first see whether the models can learn to generate according to the required scene structure. As shown in Table 1, *GNM provides almost perfect accuracy for ARROW and MNIST-4, while the baselines show significantly low performance.* It is interesting to see that for MNIST-10 all baselines completely fail while the accuracy of GNM remains high. This indicates that *the learnability of the scene structure is affected by the number of components and the dependency complexity, and GNM is more robust to this factor.* ConvDRAW with 8 steps (ConvDRAW-8) performs better than ConvDRAW with 4 steps (ConvDRAW) but still much worse than the default GNM which has 4 drawing steps. This indicates that *the hierarchical architecture and structured representation of GNM is a meaningful factor making the model efficient.* Also, from Table 2, we can see that GNM with 8 drawing steps brings further improvement. Although GENESIS is designed to learn both structured representation and density-based generation, it performs poorly in all tasks. From this, it seems that *GENESIS cannot model such scene dependency structures.*

**Discriminability.** Although the dataset allows us to evaluate the correctness of the scene structure manually, it is difficult to evaluate the clarity of the generated images manually. Thus, we use discriminability as the second metric. Note that to be realistic (i.e., difficult for the discriminator to classify), the generated image should have both correct scenes structure *and* clarity. From the result in Table 1 and Figure 4 (right), we observe a consistent result as the scene structure accuracy: *GNM samples are significantly more difficult for a discriminator to distinguish from the real images than those generated by the baselines.* The poor performance of GENESIS for this metric indicates that its generation quality is poor even if it can learn structured representation. Interestingly, GNM is more difficult to discriminate than non-structured generative models (ConvDRAWs) even if it learns the structured representation together. This, in fact, can be considered as evidence showing that *the GNM model utilizes the structured representation in such a way to generate more realistic images.*

**Log-Likelihood.** While GNM provides a better log-likelihood for the ARROW dataset than ConvDRAWs, for the MNIST datasets, ConvDRAWs perform slightly better than GNM even if the previous two metrics and the qualitative investigation clearly indicate that the ConvDRAWs provide much less realistic images than GNM. In fact, this result is not surprising but reaffirms a well-known fact that *log likelihood is not a good metric for evaluating generation quality*; as studied in [38], for high-dimensional data like our images, a high log-likelihood value does not necessarily mean a

**Table 2:** Results for ablation study. GNM-Struct is the default GNM model with StructDRAW. GNM-Gaussian uses Gaussian global prior instead of StructDraw. GNM-NoMLP removes the MLP interaction layer from StructDRAW. ConvDRAW-MLP adds an MLP interaction layer to ConvDRAW.

| Dataset | ARROW | | | MNIST-10 | | |
|---|---|---|---|---|---|---|
| Metrics | S-Acc | D-Steps | LL | S-Acc | D-Steps | LL |
| GNM-Struct | 0.976 | 11099 | 33809 | 0.824 | 2760 | 10450 |
| GNM-Gaussian | 0.784 | 10199 | 33803 | 0.096 | 1959 | 10437 |
| GNM-NoMLP | 0.656 | 8799 | 33812 | 0.128 | 2359 | 10442 |
| ConvDRAW | 0.176 | 3800 | 33740 | 0.000 | 1200 | 10544 |
| ConvDRAW-MLP | 0.844 | 2799 | 33707 | 0.104 | 1519 | 10406 |

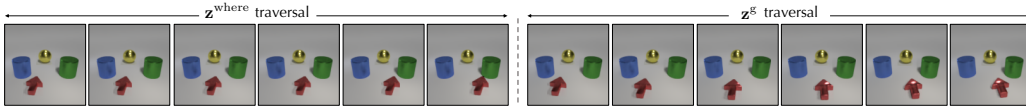

**Figure 5:** Object latent traversal and global latent traversal.

better generation quality, and vice versa. However, *the log-likelihood of GENESIS is significantly and consistently worse than the other models*.

**Ablation Study.** In Table 2, we compare various architectures to figure out the key factors making GNM outperform others. See the table caption for the description of each model. First, from the comparison between GNM-Struct and GNM-Gaussian, it seems that the *StructDRAW global prior is a key factor in GNM*. Also, by comparing GNM-Struct and GNM-NoMLP, we can see that *the interaction layer inside StructDRAW, implemented by an MLP, is also an important factor*. However, from the comparison between GNM-Struct and ConvDRAW-MLP, it seems that the MLP interaction layer is not a sole factor providing the GNM performance because, for MNIST-10, ConvDRAW-MLP still provides poor performance. Also, adding MLP interaction to ConvDRAW tends to degrade its D-Steps and LL, but it helps improve GNM. This indicates that *the hierarchical modeling and StructDRAW are the key factors realizing the performance of GNM*.

**Effects of** $\beta$**.** As the baselines in Table 1 show a very low accuracy with the default value $\beta = 1$ for the hyperparameter for KL term [23], we also tested different values of $\beta$. As shown in Figure 4, the scene structure accuracy of ConvDRAW and VAE improved as the beta value increases. However, *even for the largest value* $\beta = 10$*, the structure accuracy of ConvDRAW is still lower than (for ARROW room) or similar to (for MNIST-10) GNM with* $\beta = 1$ *while their log-likelihoods are significantly degraded*. For GNM, we only tested $\beta = [1, 2, 3]$ for MNIST-10 as it provides good and robust performance for these low values. *GNM also shows an improved structure accuracy, but a more graceful degradation of the log-likelihood is observed*.

**Novel Image Synthesis.** The dual representation of GNM, ($z^g$ for distributed representation and $z^s$ for symbolic structure), can provide an interesting way to synthesize novel scenes. As shown in Figure 5, we can generate a novel scene by controlling an object's structured representation, such as the position, independently of other components. On the other hand, we can also traverse the global distributed representation and generate images. In this case, we can see the generation also reflects the correlation between components because the arrow changes not only its position but also its pointing direction so as to keep pointing the gold ball.

## 5   Conclusion

In this paper, we proposed the Generative Neurosymbolic Machines (GNM), which combine the benefits of distributed and symbolic representation in generative latent variable models. GNM not only provides structured symbolic representations which are interpretable, modular, and compositional but also can generate images according to the density of the observed data, a crucial ability for world modeling. In experiments, we showed that the proposed model significantly outperforms the baselines in learning to generate images clearly and with complex scene structures following the density of the observed structure. Applying this model for reasoning and causal learning will be interesting future challenges. We hope that our work contributes to encouraging further advances toward combining connectionism and symbolism in deep learning.

## Broader Impact

The applicability of the proposed technology is broad and general. As a generative latent variable model that can infer a representation and also generate synthetic images, the proposed model generally shares similar effects of the VAE-based generative models. However, its ability to learn object-centric properties in an unsupervised way can help various applications requiring heavy object-centric human annotations such as various computer vision tasks. The model could also be used to synthesize a scene that can be seen as novel or fake depending on the purpose of the end-user. Although the presented model cannot generate images realistic enough to deceive humans, it may achieve this ability when combined with more powerful recent VAE models such as NVAE [39].

## Acknowledgement

SA thanks Kakao Brain and Center for Super Intelligence (CSI) for their support. The authors also thank Zhixuan Lin and the reviewers for helpful discussion and comments.

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
