[Supplementary Material]

# Supplementary Material for "Generative Neurosymbolic Machines"

## 1 Additional qualitative Results

### 1.1 Generation

In Figure 1 - 4, we show additional generation results for GNM and the baseline models. For ConvDRAW and ConvDRAW-8, we show the results for $\beta$ value 1 and 10.

### 1.2 Generation with $z^s$ resampling

In Figure 5 and 6 we show the generation results with different $z^s$ samples while the $z^g$ are fixed. In the arrow room dataset, we see the image variation is small for different $z^s$ samples though we can occasionally see some object color changes. And in the two MNIST datasets, we see some variation on the digit styles in different $z^s$ samples while the overall scene structure remains the same. Comparing the variation in Figure 5 and 6, we find the same level of certainty on the $z^s$ on both the posterior $z^g$ samples and the prior $z^g$ samples. This implies that the global representation $z^g$ is flexible enough to capture most of the information in the scene.

**Figure 1:** Additional generations results of GNM. MNIST-4 (left), MNIST-10 (middle), Arrow room (right).

**Figure 2:** Additional generations results of GENESIS. MNIST-4 (left), MNIST-10 (middle), Arrow room (right).

**Figure 3:** Additional generations results of ConvDRAW with draw steps 4 on two $\beta$ value. MNIST-4 (left), MNIST-10 (middle), Arrow room (right).

**Figure 4:** Additional generations results of ConvDRAW with draw steps 8 two different $\beta$ value. MNIST-4 (left), MNIST-10 (middle), Arrow room (right).

**Figure 5:** Results showing different $\mathbf{z}^s$ sampling while fixing $\mathbf{z}^g$ where $\mathbf{z}^g$ is inferred from the image.

**Figure 6:** Results showing different $\mathbf{z}^s$ sampling while fixing $\mathbf{z}^g$ where $\mathbf{z}^g$ is generated from the prior.

## 2 Additional experiment

### 2.1 MNIST-4-10

To evaluate GNM's ability to model more complex data variations, we generate a new task by combining datasets MNIST-4 and MNIST-10. In this setting, the model is required to model the correlation between the number of objects and the corresponding scene structure. As we can see in Figure 7, GNM can generate new scenes that reflect the ground-truth design while all baseline models fail to achieve it. This also reflects on the quantitative result shown in Table 1. We see that the generation from GNM is more difficult to distinguish from the real images and has a higher scene structure accuracy. In this task, the default GNM model with draw steps 4 has relatively lower scene accuracy than those on MNIST-4 and MNIST-10. Increasing the number of draw steps to 8 (GNM-8) significantly improves the scene accuracy. This shows that, in this task, more interaction steps are needed to model the scene structure correctly. We also test GNM and ConvDRAW with different $\beta$. Similar to the result on MNIST-4 and MNIST-10, a larger $\beta$ term brings a higher scene accuracy and a lower likelihood value. Note that GNM and GNM-8 with $\beta$ 2 still outperform ConvDRAW and ConvDRAW-8 with $\beta$ 10 in terms of scene accuracy.

### 2.2 Representation Learning

The goal of this experiment is to measure the quality of the learned structured representation. Here we use SPACE as the baseline. The results are shown in Table 2. First, we test the model's ability to infer the object position by measuring the inferred bounding boxes' average precision with the ground-truth boxes on different IoU thresholds. Second, we measure the quality of the inferred $\mathbf{z}^{\text{what}}$ representation by an object-wise classification task. More specifically, we train a two-layer MLP to classify the inferred $\mathbf{z}^{\text{what}}$ representation into 10 digit classes. The digit label of the nearest object in the dataset is used as the ground-truth lable. Both metrics are computed using the test set. As we can

**Figure 7:** Generations results for MNIST-4-10. (a) GNM, (b) GNM-8, (c) GENESIS, (d) ConvDRAW, (e) ConvDRAW-8, (f) VAE.

**Table 1:** Quantitative result on MNIST-4-10 dataset.

| Dataset | MNIST-4-10 | | |
|---|---|---|---|
| Metrics | S-Acc | D-Steps | LL |
| GNM | 0.692 | 5279 | 10693 |
| GNM-8 | 0.852 | 6639 | 10692 |
| GNM-$\beta$-2 | 0.916 | 1719 | 10684 |
| GNM-8-$\beta$-2 | 0.960 | 3599 | 10693 |
| GENESIS | 0.016 | 199 | 10084 |
| ConvDRAW | 0.000 | 959 | 10756 |
| ConvDRAW-8 | 0.004 | 1759 | 10775 |
| ConvDRAW-$\beta$-10 | 0.560 | 2399 | 10429 |
| ConvDRAW-8-$\beta$-10 | 0.880 | 1759 | 10479 |

**Table 2:** Quantitative results for representation learning.

| Model | Dataset | Avg. Precision IoU Threshold $= 0.5$ | Avg. Precision IoU Threshold $\in [0.5 : 0.05 : 0.95]$ | Classification Accuracy |
|---|---|---|---|---|
| GNM | MNIST-10 | 0.905 | 0.459 | 0.984 |
| GNM | MNIST-4 | 0.905 | 0.487 | 0.983 |
| SPACE | MNIST-10 | 0.905 | 0.453 | 0.980 |
| SPACE | MNIST-4 | 0.906 | 0.464 | 0.979 |

see, GNM and SPACE have the similar performance on the two tasks, this showcase GNM's ability to obtain good structured representations.

# 3 Auxiliary Losses and Curriculum Learning

**Auxiliary Losses**

GNM is trained by maximizing the Evidence Lower Bound (ELBO) with additional KL terms. The ELBO is shown in the following

$$\mathcal{L} = \mathbb{E}_{q_\phi(\mathbf{z}^s, \mathbf{z}^b | \mathbf{x})} \left[ \log p_\theta(\mathbf{x} \mid \mathbf{z}^s, \mathbf{z}^b) \right] - \beta_g D_{\mathrm{KL}} \left[ q_\phi(\mathbf{z}^g \mid \mathbf{x}) \parallel p_\theta(\mathbf{z}^g) \right] \tag{1}$$

$$- D_{\mathrm{KL}} \left[ q_\phi(\mathbf{z}^b \mid \mathbf{x}) \parallel p_\theta(\mathbf{z}^b \mid \mathbf{z}^g) \right] - D_{\mathrm{KL}} \left[ q_\phi(\mathbf{z}^s \mid \mathbf{x}) \parallel p_\theta(\mathbf{z}^s \mid \mathbf{z}^g) \right] . \tag{2}$$

Here, the structure representation is split into the latent structure map $\mathbf{z}^s$ and background representation $\mathbf{z}^b$. The coefficient $\beta_g$ for the KL of global representation is used in the curriculum training period and will be 1 in the remaining training stage.

Unlike the prior distribution in SPACE [6] that serve as the regularization on the posterior distributions, the structure prior of $p(\mathbf{z}^b | \mathbf{z}^g)$ and $p(\mathbf{z}^o | \mathbf{z}^g)$ in GNM are both conditional and learned from the posterior distributions. This causes the problem that by optimizing the ELBO, we cannot provide any prior knowledge to the posterior distribution to guide the inference process. To solve this problem, we introduce the following additional KL terms in the optimization objective.

$$\mathcal{L}^b = -D_{\mathrm{KL}} \left[ q(\mathbf{z}^b \mid \mathbf{x}) \parallel \mathcal{N}(0, 1) \right] \tag{3}$$

$$\mathcal{L}^o = -D_{\mathrm{KL}} \left[ q(\mathbf{z}^{\mathrm{pres}} \mid \mathbf{x}) \parallel \mathrm{Ber}(\rho) \right] - KL \left[ q(\mathbf{z}^{\mathrm{where}}, \mathbf{z}^{\mathrm{what}} \mid \mathbf{x}) \parallel \mathcal{N}(\boldsymbol{\mu}, \boldsymbol{\sigma}^2) \right] \tag{4}$$

The $\boldsymbol{\mu}$ and $\boldsymbol{\sigma}$ is further split into $\boldsymbol{\mu}_{\mathrm{what}}$, $\boldsymbol{\sigma}_{\mathrm{what}}$, and $\boldsymbol{\mu}_{\mathrm{where}}$ and $\boldsymbol{\sigma}_{\mathrm{where}}$. Here, the $\mathcal{N}(\boldsymbol{\mu}_{\mathrm{what}}, \boldsymbol{\sigma}^2_{\mathrm{what}})$ is chosen to be a standard normal distribution. The $\mathcal{N}(\boldsymbol{\mu}_{\mathrm{where}}, \boldsymbol{\sigma}^2_{\mathrm{where}})$ is chosen to encourage the bounding boxes to be tighter and closer to each grid center. The parameter $\rho$ for Bernoulli distribution is set to have a small value to encourage the model to explain the scene with as few objects as possible. With these additional KL terms, the objective function becomes the following

$$\tilde{\mathcal{L}} = \mathcal{L} + \beta_b \mathcal{L}^b + \mathcal{L}^o \tag{5}$$

Here $\beta_b$ is used for curriculum training which will be described in detail in the following section.

**Curriculum Training**

For a neural network module, modeling an individual component is usually a much simpler task than modeling a full multi-object scene. Thus, when provided multiple modules, the model should be encouraged to utilize different modules to model the individual components, e.g., modeling the foreground objects with foreground bounding boxes and the background with the background module.

However, when training GNM, we observed a different learning pattern. The model tends to explain the full scene only using the background module. We found that this is the result of the learning behavior in the early training iterations. At the initial training stage, the background module is provided more signals to optimize because, by design, it is always an activated module ($\mathbf{z}^{\mathrm{pres}} = 1$). This allows the background model to learn an accurate full scene reconstruction quickly. On the other hand, the foreground model is usually turned off ($\mathbf{z}^{\mathrm{pres}} = 0$) at the early training stage since it is under-optimized and provides rather bad object reconstructions. This again encourages the model to bias more on the background module, and, as a result, the background module dominates.

To solve this problem, we employ a curriculum learning procedure to provide more learning signal for the foreground modules in the early training iteration. First, we set its object mask for each object bounding box to occupy the full box and assign a non-zero value, e.g., 0.9, for each pixel. This forces the foreground module to be responsible for 90% of the pixel value in the boxed area. Second, $\beta_b$ is set to be 50 at the beginning and gradually annealed to 1 in 50000 steps. This limits the background capacity and thus encourages the background module to learn a simpler and more static component.

Apart from the curriculum training on the foreground module, we also perform a warm-up on the KL term of global representation. This is done by gradually increasing the value of $\beta_g$ from 0 to 1 in the first 100k steps. It allows the model to first learn a meaningful structure representation for $\mathbf{z}^s$ before optimizing the global representation that generates them.

**Figure 8:** StructDRAW Architecture. StructDRAW constructs the abstract feature map $f_L$ via multiple autoregressive steps. The output feature map $f_L$ is then used to generate the entity-based representation $\mathbf{z}^s$ to render the final generation.

## 4 Implementation Details

In this section, we describe the details of the model design. The detailed architecture is shown in Table 4 - 9. In these tables, Layer denotes the layer normalization [1] and Subconv denotes the sub-pixel convolutional layers [7].

### 4.1 GNM

**Inference Model**

The inputs to GNM are images with $128 \times 128$ resolutions. It is first provided to a convolutional neural network (CNN) to obtain a $4 \times 4$ encoding $\mathbf{f}^x$. The architecture of the image encoder is shown in Table 4. The $4 \times 4$ feature map is then used to infer the representations $\mathbf{z}^g$, $\mathbf{z}^b$, and $\mathbf{z}^s$. Here, $\mathbf{z}^s = \{\mathbf{z}^s_{hw}\}$ and $\mathbf{z}^s_{hw} = [\mathbf{z}^{\text{pres}}_{hw}, \mathbf{z}^{\text{what}}_{hw}, \mathbf{z}^{\text{where}}_{hw}, \mathbf{z}^{\text{depth}}_{hw}]$. We first use an MLP layer on top of the image encoding to infer the background representation. For the structure representations $q(\mathbf{z}^s \mid \mathbf{f}^x)$, we apply additional CNN layers to infer each of the representations $\mathbf{z}^{\text{pres}}_{hw}, \mathbf{z}^{\text{what}}_{hw}, \mathbf{z}^{\text{where}}_{hw}$, and $\mathbf{z}^{\text{depth}}_{hw}$. For the global representation, the encoding $\mathbf{f}^x$ is provided to the StructDRAW module.

**StructDRAW**

The overall architecture of StructDRAW, shown in Figure 8, is similar to ConvDRAW [3] but with two major differences: (1) it has an interaction layer that allows information mixing among the scene components, and (2) it draws on the feature level instead of the image pixel. It has 2 convolutional LSTMs (ConvLSTM) [9] for encoding and decoding. The input to the encoder ConvLSTM is a concatenation of 3 components, the image encoding $\mathbf{f}^x$, the hidden state previous-step decoder $\mathbf{h}_{\text{dec},\ell-1}$, and the element-wise difference between the accumulated decoding and the image encoding $\mathbf{f}^x - \mathbf{f}_{\ell-1}$. Finally, a 3-layer MLP is used as the interaction layer to compute the posterior parameter of $q(\mathbf{z}^g_\ell \mid x)$ on the current step.

When generating an image, the StructDRAW module draws a structure feature map $\mathbf{f}$ by sampling the $\mathbf{z}_\ell^g$ auto-regressively. This is done as the following. On each step of the drawing, the ConvLSTM takes the previous feature map $\mathbf{f}_{\ell-1}$ and the current global representation $\mathbf{z}_\ell^g$ as input and update its hidden state $\mathbf{h}_{\text{dec},\ell}$. Here, an MLP decoder is used to decode the $\mathbf{z}_\ell^g$ into a feature map. The output feature map for the current time step is obtained by $\mathbf{f}_\ell = \sum_{l=1}^{\ell} \text{CNN}(\mathbf{h}_{\text{dec},l})$. Here the function CNN is a single-layer convolutional network. Given the global representation, the background representation is obtained using an MLP layer that takes the concatenation of $\mathbf{z}_\ell^g$ at every step as input. The overall algorithm of StructDRAW is illustrated in Algorithm 1.

**Rendering**

The output feature map from StructDRAW is used to generation the symbolic representation map $p(\mathbf{z}^s \mid \mathbf{f})$. For all of our generation samples, we directly take the mode of $\mathbf{z}^s$ instead of sampling from the prior distribution. Note that we share the parameter of the network $q(\mathbf{z}^s \mid \mathbf{f}^x)$ and $p(\mathbf{z}^s \mid \mathbf{f})$. This encourages the model to generate the structure feature map that is consistent with the input feature during training.

Given the symbolic representation map $\mathbf{z}^s$, the rendering process is similar to that in SPACE [6] and SCALOR [4]. For each object in the foreground, we first obtain its RGB appearance $\mathbf{o}_{hw}$ and segmentation mask $\mathbf{m}_{hw}$ by decoding from the $\mathbf{z}^{\text{what}}$ representation using a CNN decoder, which is shown in Table 5. The full foreground mask $\mathbf{M}$ is then obtained by summing all object masks into a full image using the spatial transformer network (STN) using $\mathbf{z}^{\text{where}}$. Similarly, each object image $\mathbf{o}_i$ is mapped into the full-image size and gives $\mathbf{x}_i$. To determine which object should be drawn in a foreground pixel position (when multiple objects occupy the pixel), we first compute the responsibility $\boldsymbol{\gamma}_i$ using $\mathbf{z}^{\text{pres}}$, $\mathbf{z}^{\text{depth}}$, and $\mathbf{z}^{\text{where}}$ and then the full foreground image $\mathbf{x}^{\text{fg}}$ is obtained by multiplying the object images with the normalized responsibilities. The background image $\mathbf{x}^b$ is generated by a background decoder shown in Table 6. The final image is then computed by $\mathbf{x} = \mathbf{x}^{\text{fg}} + (1 - \mathbf{M}) \odot \mathbf{x}^b$. The full rendering process is illustrated in Algorithm 2.

Table 7 describes the rest of the network structures that are not specified by Table 4 - 6. Note that in Table 7, all convolutional and MLP layers except the output layers are followed by a CELU activation function [2] and a layer normalization [1].

## 4.2 Baseline models

Our implementation of GENESIS is based on the official PyTorch implementation. We found that with the default setting in official code, we are unable to make GENESIS decompose the scene into components. Instead, the model tends to cluster the objects into components base on their colors or locations. Thus, to encourage a correct decomposition and generation, we make the following modifications on the official code: (a) we design a learning rate schedule where it starts with a higher value and reset to a lower one in a few thousand steps. (b) instead of optimizing the Constrained Optimisation objective (GECO) [5], we optimize the evidence lower bound with a $\beta$ value of 15 for arrow room dataset and 10 for MNIST dataset. (3) we reduce the number of layers for the spatial broadcast decoder [8] from 4 to 3. We found that the modifications allow GENESIS to decompose the scene correctly while also improve its generation quality on three datasets.

For VAE and ConvDRAW, we use our own implementation. We first use an image encoder to obtain an image encoding for both models. Its architecture is designed to have the same structure as the image encoder in GNM, shown in Table 4. Then for VAE, we use a 2-layer CNN with filter sizes of [128, 128], and kernel sizes of [3, 4] on top of the image encoding to compute the parameter for the latent representation. For ConvDRAW, we apply a similar architecture of StructDRAW on top of the image encoding shown in Table 7, while the interaction MLP is replaced with a 2-layer CNN with filter sizes of [128, 64], and kernel sizes of 3. The architecture of the image decoders for the two models are shown in Table 8 and 9.

Table 3 shows the model size each model used for the three datasets

**Table 3:** Model size comparison.

| Dataset | ARROW | MNIST-10 | MNIST-4 |
|---|---|---|---|
| GNM | 8.6M | 2.9M | 2.9M |
| GENESIS | 13.9M | 13.9M | 13.9M |
| ConvDRAW | 4.7M | 1.5M | 1.5M |
| VAE | 1.5M | 1.5M | 1.5M |

---

**Algorithm 1** StructDRAW

---

**Input:** Image encoding $\mathbf{f}^x$
**Output:** Global representation $\mathbf{z}^g$, structure feature map $\mathbf{f}$,
　　　　Prior distribution $\{p(\mathbf{z}^g_\ell)\}_\ell$, Posterior distribution $\{q(\mathbf{z}^g_\ell)\}_\ell$
// Initialize hidden state and feature map
$\mathbf{h}_{\text{enc},0}, \mathbf{h}_{\text{dec},0}, \mathbf{f}_0 = \text{init\_zeros}()$
// StructDRAW
**for** $\ell \leftarrow 1$ **to** $L$ **do**
$\quad \boldsymbol{\mu}_{p,\ell}, \boldsymbol{\sigma}_{p,\ell} = \text{MLP}^{\text{int}}_{\text{dec}}(\mathbf{h}_{\text{dec},\ell-1})$
$\quad p(\mathbf{z}^g_\ell) = \mathcal{N}(\boldsymbol{\mu}_{p,\ell}, \boldsymbol{\sigma}_{p,\ell})$
$\quad$ **if** *is_inference* **then**
$\quad\quad \mathbf{h}_{\text{enc},\ell} = \text{ConvLSTM}_{\text{enc}}(\mathbf{h}_{\text{enc},\ell-1}, \text{CAT}[\mathbf{h}_{\text{dec},\ell-1}, \mathbf{f}^x, \mathbf{f}^x - \mathbf{f}_{\ell-1}])$
$\quad\quad \boldsymbol{\mu}_{q,\ell}, \boldsymbol{\sigma}_{q,\ell} = \text{MLP}^{\text{int}}_{\text{enc}}(\mathbf{h}_{\text{enc},\ell})$
$\quad\quad q(\mathbf{z}^g_\ell) = \mathcal{N}(\boldsymbol{\mu}_{q,\ell}, \boldsymbol{\sigma}_{q,\ell})$
$\quad\quad \mathbf{z}^g_\ell \sim q(\mathbf{z}^g_\ell)$
$\quad$ **else**
$\quad\quad \mathbf{z}^g_\ell \sim p(\mathbf{z}^g_\ell)$
$\quad$ **end**
$\quad \mathbf{d}_\ell = \text{MLP}^g_{\text{dec}}(\mathbf{z}^g_\ell)$
$\quad \mathbf{h}_{\text{dec},\ell} = \text{ConvLSTM}_{\text{dec}}(\mathbf{h}_{\text{dec},\ell-1}, \mathbf{d}_\ell)$
$\quad \mathbf{f}_\ell = \mathbf{f}_{\ell-1} + \text{CNN}_{\text{out}}(\mathbf{h}_{\text{dec},\ell})$
**end**
$\mathbf{z}^g = \text{CAT}[\{\mathbf{z}^g_\ell\}_l]$
**if** *is_inference* **then**
$\quad$ **Output:** $\mathbf{z}^g, \mathbf{f}, \{p(\mathbf{z}^g_\ell)\}_\ell, \{q(\mathbf{z}^g_\ell)\}_\ell$
**else**
$\quad$ **Output:** $\mathbf{z}^g, \mathbf{f}, \{p(\mathbf{z}^g_\ell)\}_\ell$
**end**

---

**Table 4:** Image Encoder

| Layer | Size/Ch. | Stride | Norm./Act. |
|---|---|---|---|
| Input | 128(3d) | | |
| Conv $4 \times 4$ | 16 | 2 | Layer/CELU |
| Conv $3 \times 3$ | 16 | 1 | Layer/CELU |
| Conv $4 \times 4$ | 32 | 2 | Layer/CELU |
| Conv $3 \times 3$ | 32 | 1 | Layer/CELU |
| Conv $4 \times 4$ | 64 | 2 | Layer/CELU |
| Conv $3 \times 3$ | 64 | 1 | Layer/CELU |
| Conv $4 \times 4$ | 128 | 2 | Layer/CELU |
| Conv $3 \times 3$ | 128 | 1 | Layer/CELU |
| Conv $4 \times 4$ | 128 | 2 | |

---
**Algorithm 2** Rendering
---
**Input:** Structure representation $\{\mathbf{z}_{hw}^{\text{pres}}, \mathbf{z}_{hw}^{\text{what}}, \mathbf{z}_{hw}^{\text{where}}, \mathbf{z}_{hw}^{\text{depth}}\}$, background representation $\mathbf{z}^b$
**Output:** Image reconstruction $\tilde{\mathbf{x}}$
// Obtain the object appearance $\mathbf{o}_{hw}$ and segmentation mask $\mathbf{m}_{hw}$
$\mathbf{o}_{hw}, \mathbf{m}_{hw} = \text{GlimpseDecoder}(\mathbf{z}_{hw}^{\text{what}})$
// Obtain the background $\mathbf{x}^b$
$\mathbf{x}^b = \text{BgDecoder}(\mathbf{z}^b)$
// Foreground object rendering
**for** $i \leftarrow 1$ **to** $HW$ **do**
$\quad \mathbf{x}_i^{\text{fg}} = \text{STN}^{-1}(\mathbf{o}_i, \mathbf{z}_i^{\text{where}})$
$\quad \boldsymbol{\gamma}_i = \text{STN}^{-1}(\mathbf{m}_i \cdot z_i^{\text{pres}} \cdot \sigma(-\mathbf{z}_i^{\text{depth}}), \mathbf{z}_i^{\text{where}})$
$\quad \boldsymbol{\gamma}_i = \text{normalize}(\boldsymbol{\gamma}_i, \forall i)$
**end**
$\mathbf{x}^{\text{fg}} = \sum_i \mathbf{x}_i^{\text{fg}} \boldsymbol{\gamma}_i$
// Foreground mask rendering
**for** $i \leftarrow 1$ **to** $HW$ **do**
$\quad \mathbf{M}_i = \text{STN}^{-1}(\mathbf{m}_i, \mathbf{z}_i^{\text{where}})$
**end**
$\mathbf{M} = \min(\sum_i \mathbf{M}_i, 1)$
// Foreground background combination
$\tilde{\mathbf{x}} = \mathbf{x}^{\text{fg}} + (1 - \mathbf{M}) \odot \mathbf{x}^b$
**Output:** $\tilde{\mathbf{x}}$
---

**Table 5:** Object patches decoder

| Layer | Size/Ch. | Stride | Norm./Act. |
|---|---|---|---|
| Input | 1(64d) | | |
| Subconv $3 \times 3$ | 128 | 2 | Layer/CELU |
| Subconv $3 \times 3$ | 64 | 2 | Layer/CELU |
| Subconv $3 \times 3$ | 32 | 2 | Layer/CELU |
| Subconv $3 \times 3$ | 16 | 2 | Layer/CELU |
| Subconv $3 \times 3$ | 8 | 2 | Layer/CELU |
| Subconv $3 \times 3$ | 4 | 2 | |
| Sigmoid | | | |

**Table 6:** Background decoder

| Layer | Size/Ch. | Stride | Norm./Act. |
|---|---|---|---|
| Input | 1(10d) | | |
| Subconv $1 \times 1$ | 128 | 4 | Layer/CELU |
| Subconv $1 \times 1$ | 64 | 2 | Layer/CELU |
| Subconv $1 \times 1$ | 32 | 4 | Layer/CELU |
| Subconv $1 \times 1$ | 16 | 2 | Layer/CELU |
| Subconv $1 \times 1$ | 8 | 2 | Layer/CELU |
| Subconv $3 \times 3$ | 4 | 1 | |
| Sigmoid | | | |

**Table 7:** Additional network architecture

| Description | Symbol | Structure |
|---|---|---|
| Encoder ConvLSTM | $\text{ConvLSTM}_{\text{enc}}$ | ConvLSTM(128, kernel_size=3, stride=1) |
| Decoder ConvLSTM | $\text{ConvLSTM}_{\text{dec}}$ | ConvLSTM(128, kernel_size=3, stride=1) |
| Compute $\mathbf{z}^s$ from $\mathbf{f}^x$ | $q(\mathbf{z}^s \mid \mathbf{f}^x)$ | StackConv([128, 128, 70]) |
| Structure interaction network | $\text{MLP}_{\text{enc}}^{\text{int}}, \text{MLP}_{\text{dec}}^{\text{int}}$ | MLP([512, 512, 64]) |
| StructDRAW output CNN | $\text{CNN}_{\text{out}}$ | Conv(128, kernel_size=3, stride=1) |
| $\mathbf{z}_\ell^g$ decoder | $\text{MLP}_{\text{dec}}^g$ | MLP([512, 1024, 2048]) |
| Background inference network | $q(\mathbf{z}^b \mid \mathbf{f}^x)$ | MLP([512, 256, 20]) |
| Background generation network | $p(\mathbf{z}^b \mid \mathbf{z}^g)$ | MLP([128, 64, 20]) |

**Table 8:** Architecture of VAE decoder

| Layer | Size/Ch. | Stride | Norm./Act. |
|---|---|---|---|
| Input | 1(128d) | | |
| Subconv $1 \times 1$ | 128 | 4 | Layer/ReLU |
| Subconv $3 \times 3$ | 128 | 1 | Layer/ReLU |
| Subconv $1 \times 1$ | 64 | 2 | Layer/ReLU |
| Subconv $3 \times 3$ | 64 | 1 | Layer/ReLU |
| Subconv $1 \times 1$ | 32 | 2 | Layer/ReLU |
| Subconv $3 \times 3$ | 32 | 1 | Layer/ReLU |
| Subconv $1 \times 1$ | 16 | 2 | Layer/ReLU |
| Subconv $3 \times 3$ | 16 | 1 | Layer/ReLU |
| Subconv $1 \times 1$ | 16 | 2 | Layer/ReLU |
| Subconv $3 \times 3$ | 16 | 1 | Layer/ReLU |
| Subconv $1 \times 1$ | 8 | 2 | Layer/ReLU |
| Subconv $3 \times 3$ | 3 | 1 | |
| Sigmoid | | | |

**Table 9:** Architecture of ConvDRAW decoder

| Layer | Size/Ch. | Stride | Norm./Act. |
|---|---|---|---|
| Input | 4(128d) | | |
| Subconv $3 \times 3$ | 128 | 1 | Layer/ReLU |
| Subconv $1 \times 1$ | 64 | 2 | Layer/ReLU |
| Subconv $3 \times 3$ | 64 | 1 | Layer/ReLU |
| Subconv $1 \times 1$ | 32 | 2 | Layer/ReLU |
| Subconv $3 \times 3$ | 32 | 1 | Layer/ReLU |
| Subconv $1 \times 1$ | 16 | 2 | Layer/ReLU |
| Subconv $3 \times 3$ | 16 | 1 | Layer/ReLU |
| Subconv $1 \times 1$ | 16 | 2 | Layer/ReLU |
| Subconv $3 \times 3$ | 16 | 1 | Layer/ReLU |
| Subconv $1 \times 1$ | 8 | 2 | Layer/ReLU |
| Subconv $3 \times 3$ | 3 | 1 | |
| Sigmoid | | | |