[Reviews · NeurIPS 2020]

Review 1

Summary and Contributions: Update: I've read the rebuttal and it addressed my concerns. I'm increasing my score to 8. ### This paper introduces a novel hierarchical VAE, which allows generating visual scenes compositionally in a per-component basis. This is achieved by using a hierarchical global prior similar to the DRAW (Gregor et. al, "Towards Conceptual Compression") model, which then conditions lower-level structural latent variables. The final scene is composed of components, that are independently rendered from the structured latent variables, and an independently-generated background. The model is similar to the likes of AIR (Eslami et. al.), SPACE (Lin et. al) and GENSIS (Engelcke et. al.), but shows superior performance in density estimation, the quality of generated samples, and the ability to learn the global scene structure. The model is evaluated on two variants of the multi-MNIST dataset, and a dataset similar to CLEVR. The datasets are designed to test the ability of learning global scene structure from data, and the authors report relevant metrics: manual evaluation of generated samples with respect to their structure, the difficulty of classifying samples as ground-truth or generated with a separately trained classifier.

Strengths: - The introduced model is very close to several previously introduced component-wise generative models, but what makes it different is the presence of a hierarchical global prior which conditions local latent variables. This is novel, and the strong empirical evaluations do corroborate the benefits of such a design. - Empirical evaluations are well designed, and they try to capture how well does the model capture both local and global image statistics. I have not seen many such evaluation attempts in prior art, or at least not to this extent. Well done! - The model is theoretically sound and is expressed in the VAE framework. The ELBO and generative and inference models are well defined. - There are ablation studies that remove all novel components from the model, and show that both the hierarchical prior and an "interaction MLP" are necessary for optimal model performance. - This work has high potential significance, as there is quite some interesting in the community in this topic: see e.g. the recent Object-Oriented Learning workshop at ICML this year.

Weaknesses: - The paper tries to draw a distinction between models with purely distributed and purely symbolic representation. However, in doing so, the authors seem to forget that models like AIR and SPACE use symbolic representations very conservatively, and the majority of their representation is still distributed and therefore groundable (as per terminology introduced in the paper). In fact, I am not aware of any purely-symbolic component-wise generative models, other than maybe some works from Josh Tenenbaum's group. - Even though the current work mentions SPACE repeatedly, the introduced model is not evaluated against SPACE, which is perhaps the best component-based genenerative model on the market. - The authors do not report model sizes (in terms of the number of parameters), nor specific hyperparameters used to instantiate or train the models, nor do they show training curves. While I expect that considered model had at least similar sizes (same order of magnitude, hopefully), it is not clear if increasing model size of e.g. GENESIS could lead to results competitive with the current model. - While the designed datasets are nice, there is no evaluation on any real-world data, or even datasets used by other works. Evaluating on such datasets, e.g. CLEVR, would allow to directly compare numbers between different papers, therefore allowing a reader to judge the quality of the implementations used in the current work. This is significant, as we have also done some experiments with GENSIS, and we were able to get digit-wise decompositions on multi-MNIST datasets, which seems to be not the case in the reported results. Why is that? - The claims of the paper are somewhat strong. Straight in the abstract, we read that ALL prior methods for component-wise generation have failed to learn the true data density. Now, this might be the case on your data, but I remember seeing some convincing samples in the respective papers. Please tone it down. - The paper uses a variant of DRAW for the global latent variable, but does not discuss other modelling possibilities that would be at least as powerful. For example, one could perform semi-amortized iterative inference or use an autoregressive model.

Correctness: Both theory and evaluation seem to be correct.

Clarity: The paper is in an ok state, but some things could be made more clear: - L35-40: the distinction between the ability to generate and to synthesize by latent traversals is unclear. I am actually not sure how they are different? - Fig 1d) is a bit complicated; it would be nice if you could have a separate figure for DRAW, explained how it works, and then move on to the full model. - in Section 2, it seems to me that "grounding" is simply "learning", where learning can be done, and "inference" is the forward pass of an amortized-inference model. Is this correct? if yes, then perhaps it would be easier to understand if you first describe how things are (in terms of learning and forward-pass), and introduce your own terminology only after. As it stands, I needed to spend some time to associate the new concepts with what I already know. L99 - "this ability is not properly studied" is somewhat harsh. I thought it was fine in the context of that paper. L131: the s\z notation is a bit surprising, as no set notation was previously introduced. - eq2: in the RHS, p_\theta (z_{hw}^s|z^g) should have "hw" as an index directly in p (p_{hw}) or it should be used as a part of the conditioning set; the current notation is incorrect. - The model uses an "interaction MLP", but I have no idea what it is, how it is used, or how it is added to DRAW in the ablation experiment. Why is it called interaction? What are the inputs and outputs? This is very unclear. - The paper mentions that global and local latents form a "dual" representation. I do not understand what "dual" should mean here? They encode different things. - L298-305: you study the influence of the beta parameter, but it was not mentioned ever before. Why is this necessary?

Relation to Prior Work: The work very clearly discusses connection to prior art.

Reproducibility: No

Additional Feedback: - The model is interesting enough, that I think evaluation in non-component-wise generative modelling is warranted--that is, I would like to see how well such latent-space DRAW does when compared to DRAW and other large-scale generative models on large-scale datasets like ImageNet. - The global and local latent variables are inferred independently. While the results show that it does work fine, it seems a little disappointing. If this is the case, one could have used a non-hierarchical prior with flow, or perhaps not use hierarchy and use flows all the way. - It would be nice to see images generated by resampling the local latents while the global one is fixed, both in the case where the global is sampled from the prior and when it is inferred from an existing image. - in Section 3.3, you use additional KL terms to regularize the learnt representation and enforce some behaviour. Could this be reinterpreted as having priors that are mixtures of the learned priors and the fixed ones? - In the "broder impact" section you say that VAEs cannot generate good-enough samples to be dangerous. Well, see Vahdat et. al., "NVAE: A Deep Hierarchical Variational Autoencoder".


Review 2

Summary and Contributions: The authors introduce Generative Neurosymbolic Machines (GNM) a latent variable model for structured representation learning that can carry out density-based generation. This ability is a result of the model’s hierarchical structure that consists of a global unstructured latent variable and symbolic structured latent variable. The authors provide experimental results on three visual datasets and analyse the performance of GNMs and a number of other relevant baselines (GENESIS, ConvDRAW and VAEs) using several metrics.

Strengths: - The paper is clearly written and easy to follow. - The model is well motivated and the authors place their model well into the context of current related models. - The results look promising and convincing and although they are relatively toyish they consider a range of baselines and metrics and overall carry them out thoroughly.

Weaknesses: - Overall I think there are no major flaws. I think it would be interesting for the authors to elaborate on why they have chosen the particular form for the symbolic representations (why that shape and how does the size limit the representation? Does each of the z’s map to something?). Since the argument is that those representations are more interpretable it would be helpful to flesh that out more to make the case. This could also benefit from some experiments, visualising the analysis of these symbolic latents (beyond the traversal in Figure 5, which is already really nice).

Correctness: The methodology seems correct.

Clarity: Yes.

Relation to Prior Work: Yes, the authors mention the relevant related models and discuss how they differ from their approach. They also include some of these related models as baselines in their experiments.

Reproducibility: Yes

Additional Feedback: Questions and comments: - If there are a fixed number of objects, what is the idea behind modelling ‘what’ with a Gaussian rather than a categorical, for example? - The fact that the results show that the Log-likelihood is indeed not a good measure of image reconstruction is interesting. - The latent traversal example whereby the traversal of ‘where’ alone does not capture the relation of the arrow is really cool. Small nitpicks: - In Figure 1 ‘Grounding’ in misspelled. - In the last paragraph of the introduction I would not write ‘we propose our proposed method’.


Review 3

Summary and Contributions: This paper proposes a generative model that combines structured (object-centric) representations with with a global context. This so called Generative Neurosymbolic Machine (GNM) is shown to be able to both learn object-structured representations, and to generate samples that reflect the global structure of the input. GNMs are are based on the recent SPACE model, and extend it with a global context that is able to capture the relations between objects in a scene. The resulting model is a hierarchical latent variable model where the global context generates the spatially organized object representations using an autoregressive generative process akin to DRAW, but in latent space instead of pixel space.

Strengths: The problem of capturing object structure on the one hand, while also capturing global structure (ie. relations between objects) is an important research question, and of great relevance to the community. The proposed method is sound and has a solid theoretical basis. The paper also introduces several variants of known datasets, that have been specifically designed to evaluate the ability of GNMs to capture the global structure of a scene. The generated images look convincing and the paper clearly demonstrates that GNMs improve upon several relevant baselines and is better able to capture the desired structure.

Weaknesses: While the paper clearly demonstrates the generative capacities of GNMs I think it is missing proper evaluation of their structured representation learning ability. Only the qualitative evaluations of Figure 3 and Figure 5 allow any insights into the learned inner structure of the model. Since this ability is an important part of the motivation of this paper, it should be properly evaluated too. I.e. by measuring bounding box IOU, or using the object representations to predict ground truth factors.

Correctness: Generative models are difficult to evaluate. But the authors have found two ways of evaluating the results: Manual investigation of the generated structure, and measuring how long it takes to train a discriminator to 90% accuracy. If I understand correctly, the authors have performed the manual checking themselves. In this case they should publish the corresponding data, so that the results can be checked by others. Also for Arrow Room an established metric like FID score might have been a better choice to evaluate image quality, though it seems unlikely that this would have affected any of the results.

Clarity: The paper is well written and easy to understand. I also did like the discussion of grounding in the beginning.

Relation to Prior Work: Since object-centric generation scenes is central to this paper, it should also reference and discuss [1] [1] van Steenkiste, Sjoerd, et al. "Investigating object compositionality in generative adversarial networks." Neural Networks (2020).

Reproducibility: Yes

Additional Feedback:

[Author Response · NeurIPS 2020]

We thank reviewers for their insightful and positive feedback! We are encouraged that they find our motivation and idea novel (R1), important (R1, R4), of value to the community (R1, R4), and has high potential significance (R1). We are glad they found our paper does not have major flaw (R2), well written (R2, R4), theoretically sound (R1, R4), our empirical evaluation well designed and thoroughly performed (R1, R2, R4), and the experimental results promising (R1, R2). We address the reviewer's comments below and will incorporate all feedback.

**[R1] GNM distinguishes between purely symbolic and purely distributed representation while AIR and SPACE use it conservatively:** Not true. GNM does not require purely symbolic representations. In the structured representation layer, we are actually using hybrid (symbolic +distributed) representation which is exactly the same as SPACE and AIR, i.e., distributed for $z^{\text{what}}$ and symbolic for the others. Yes, the distributed part is still groundable. We will make this point clearer. **Comparison to SPACE is missing:** SPACE uses independent priors for symbolic priors and thus *by design* cannot model the distribution of global scene structure. See also a relevant answer for R4. **Reporting model sizes and hyperparameters:** Yes, we use comparable model sizes. We will include the model sizes, hyperparameters, and training curve in the supplementary material. We will also release our code. **Increasing model size of GENESIS would make it better?** In our experiments, GENESIS used 4 times more parameters (14M) than our model (3M). This is because GENESIS uses additional conv-layers followed by MLP layers for the encoder and decoder of the segmentation networks which our model does not need. **Why not compare to existing datasets, e.g. CLEVR:** We agree that to compare on existing datasets can be better if possible. However, we found that objects in existing datasets are rather independently generated without complex spatial and inter-object relations and thus do not reveal required global scene structures. This makes it difficult to evaluate the learned global structure, which is of our main interest. For follow-up works, we will release our datasets. **Why GENESIS cannot obtain digit-wise decomposition for multi-MNIST scenes:** Although we have worked hard to make GENESIS decompose the multi-MNIST scene, we could not observe the decomposition for this dataset. We believe that our result is valid. First, similar to our result, the results in Figure 3 and Figure 4 in the GENESIS paper, exhibit strong color-based segmentation bias (e.g., the circles in the wall). We actually also observed that in our Arrow-Room dataset, GENESIS often grouped different objects into the same segmentation when they have the same color. A careful hyperparameter tuning was required to prevent this. We believe that this is a reasonable thing to observe in GENESIS because it does not use the *locality* principle of objectness. Second, it is indeed interesting to know that R1 has observed the decomposition on a similar dataset. It would have been great if we were able to access the result to compare with ours. However, given that there is no published result about the multi-MNIST task using GENESIS including R1's experience, our result is the first and the only accessible result on which we can make any reasonable decision. **Strong claim that all prior methods have failed:** We fully agree and thanks for pointing this. We will tone down this and the sentence "this ability is not properly studied". **Other modeling possibilities:** We agree that it is indeed an interesting suggestion. The SPACE paper shows that the auto-regressive approach does not scale gracefully and iterative inference (e.g., of IODINE) is known to be computationally expensive. However, as pointed, we note that our proposed framework is more generally applicable beyond the SPACE representation style. **Interaction MLP** is a standard MLP that takes the CNN feature map as input and mixes (interacts) globally. Lacking this, standard ConvDRAW do not mix information between features spatially distant. **Dual representation:** both $z^s$ and $z^g$ are inferred from the same input $x$. So they are different representations encoding the same information. **Beta parameter effect** is studied because some VAE-family models (ConvDRAW in our case) work better with a different beta. We thank all the **additional feedback** which will make our paper clearer.

**[R2] Why the particular form of the symbolic representation:** In this work, we have taken, as an example implementation of the symbolic inference model $q(z^s|x)$, the SPACE model, and thus uses the same symbolic representation of theirs. However, the proposed framework is more general and not limited to the way SPACE represents an object. It can be applied to make any symbolic or hybrid (symbol + distributed) representation conform a proper distribution. **More visualization would be helpful although Figure 5 is "already really nice":** We thank you for suggesting this. We will add more visualization if we come up with an additional idea for that. **Why not categorical variable for $z^{\text{what}}$:** because the categorical value cannot model within-category variation.

**[R4] Need the evaluation of structured representation learning ability.** Thanks for suggesting this. We initially thought that we do not need this experiment because our structured representation learning is basically the same as SPACE (with a single segment for the background modeling). However, following the suggestion, we performed the suggested experiments by comparing ours and SPACE. We indeed obtained a similar performance between ours and SPACE. Specifically, for average precision IOU on the MNIST-10 dataset, we obtained 0.459 and 0.453 for our model and SPACE, respectively. And for classification accuracy on the recognized $z^{\text{what}}$ representation, we obtained 0.984 and 0.980 for ours and SPACE, respectively. The SPACE paper demonstrates its advantages against MoNET and IODINE in some settings and without loss of generality, we can extend that result to our model. We will add this result to the revision. **Manual checking data:** As suggested, we will publish the corresponding data we used for the manual investigation. **FID Score:** We will consider adding FID score. However, as agreed, it may not affect the main result of the paper. **Missing reference:** Thanks! We will add the suggested paper.

[Meta-Review · NeurIPS 2020]

Three knowledgeable referees support accept and I accept. We encourage and expect the authors to incorporate the reviewers' suggestions for improving the paper.